# Siamese Masked Autoencoders

**Agrim Gupta**[1*]    **Jiajun Wu**[1]    **Jia Deng**[2]    **Li Fei-Fei**[1]
[1]Stanford University, [2]Princeton University

## Abstract

Establishing correspondence between images or scenes is a significant challenge in computer vision, especially given occlusions, viewpoint changes, and varying object appearances. In this paper, we present Siamese Masked Autoencoders (SiamMAE), a simple extension of Masked Autoencoders (MAE) for learning visual correspondence from videos. SiamMAE operates on pairs of randomly sampled video frames and asymmetrically masks them. These frames are processed independently by an encoder network, and a decoder composed of a sequence of cross-attention layers is tasked with predicting the missing patches in the future frame. By masking a large fraction ($95\%$) of patches in the future frame while leaving the past frame unchanged, SiamMAE encourages the network to focus on object motion and learn object-centric representations. Despite its conceptual simplicity, features learned via SiamMAE outperform state-of-the-art self-supervised methods on video object segmentation, pose keypoint propagation, and semantic part propagation tasks. SiamMAE achieves competitive results without relying on data augmentation, handcrafted tracking-based pretext tasks, or other techniques to prevent representational collapse.

## 1 Introduction

*"The distinction between the past, present, and future is only a stubbornly persistent illusion."*

—Albert Einstein

Time is a special dimension in the context of visual learning, providing the structure within which sequential events are perceived, cause-effect relationships are learned, objects are tracked as they move through space, and future events are predicted. Central to all of these capabilities is the ability to establish visual correspondence over time. Our visual system is adept at establishing correspondence between scenes despite occlusions, viewpoint changes, and object transformations. This capability is *unsupervised*, critical to human visual perception, and remains a significant challenge in computer vision. Equipping machines with such a capability enables a wide range of applications such as object segmentation and tracking in videos, depth and optical flow estimation, and 3D reconstruction [1–8].

A powerful self-supervised learning paradigm is predictive learning, i.e., predicting any unobserved or hidden part of the signal from any observed or unhidden part of the signal [9]. Notably, this form of predictive learning has been used for learning correspondences [10–12] by predicting the colors of grayscale future frame by observing a (colorful) past reference frame. However, the performance of these methods has trailed behind contrastive self-supervised learning [13] approaches. State-of-the-art methods [14–17] for learning correspondence primarily employ some form of contrastive learning [13]. Intuitively, contrastive learning-based approaches are well-suited for the task of learning correspondence, as they utilize extensive data augmentation to learn features invariant to changes in pose, lighting, viewpoint, and other factors. However, a major criticism of contrastive approaches is their reliance on careful selection of augmentations to learn useful invariances [18], along with a suite of additional components [19, 20, 17, 21] to prevent representational collapse.

---

[*]Correspondence to `agrim@stanford.edu`

37th Conference on Neural Information Processing Systems (NeurIPS 2023).

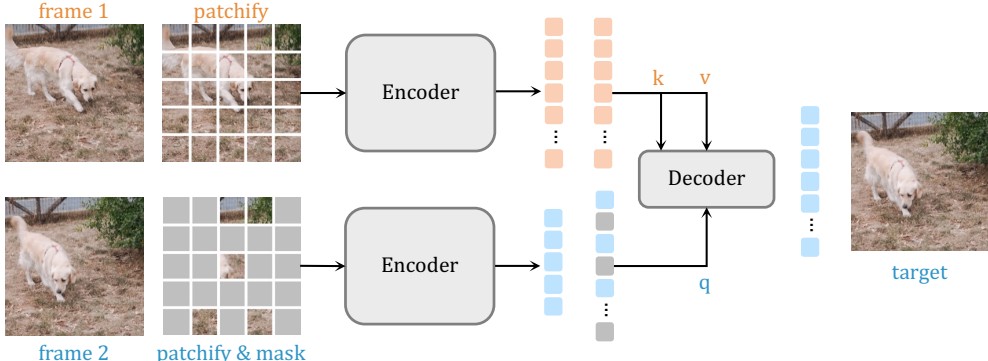

Figure 1: **Siamese Masked Autoencoders.** During pre-training we randomly sample a pair of video frames and randomly mask a huge fraction (95%) of patches of the future frame while leaving the past frame unchanged. The two frames are processed *independently* by a siamese encoder parametrized by a ViT [31]. The decoder consists of a sequence of cross-attention layers and predicts missing patches in the future frame. Videos available at this project page.

Recently, predictive learning methods like masked language modelling [22, 23] and masked visual modeling (MVM) [24–26] have demonstrated promising results in natural language processing and computer vision domains. MVM methods like Masked Autoencoders (MAE) learn good visual representations without relying on data augmentation by learning to reconstruct the missing patches from randomly masked input image patches. Extending MVM methods from images to videos for learning correspondence is however nontrivial for two reasons. First, features learned by MAEs are specialized for the pixel reconstruction task, which show excellent downstream performance on finetuning, but do not transfer well in zero-shot settings. Second, existing extensions of MAEs in the video domain [27, 28] also symmetrically mask a huge fraction of patches across all frames. Unlike images, which are (approximately) isotropic [29], the temporal dimension is special [30], and not all spatio-temporal orientations are equally likely. Hence, symmetrically treating spatial and temporal information might be sub-optimal. Indeed, MAEs trained on videos do not outperform MAEs trained on ImageNet on video instance tracking benchmarks (Table 1).

To address these limitations, we present Siamese Masked Autoencoders (SiamMAE): a simple extension of MAEs for learning visual correspondence from videos. In our approach, two frames are randomly selected from a video clip, with the future frame having a significant portion (95%) of its patches randomly masked, while the past frame is left intact. These frames are processed *independently* by an encoder network, and a decoder composed of a sequence of cross-attention layers is tasked with predicting the missing patches in the future frame. Our *asymmetric* masking approach encourages the network to model object motion, or in other words, to understand *what went where* [32]. Simple extensions of MAEs to frames with symmetric masking wastes model capacity on modeling low-level image details. However, by providing the entire past frame as input, our network is primarily focused on *propagating* the patches from the past frame to their corresponding locations in the future frame. The cross-attention layers in our decoder serve a function akin to the affinity matrix often employed in self-supervised correspondence learning approaches. Empirically, we find that the combination of asymmetric masking, a siamese encoder, and our decoder can effectively learn features suitable for tasks requiring fine-grained and object-level correspondence.

Despite the conceptual simplicity of our method, it outperforms state-of-the-art self-supervised methods on video object segmentation, pose keypoint propagation, and semantic part propagation. Moreover, our ViT-S/16 models significantly outperform larger models trained on ImageNet (+8.5% $\mathcal{J}\&\mathcal{F}_m$ for ViT-B) and Kinetics-400 (+7.4% $\mathcal{J}\&\mathcal{F}_m$ for ViT-L) via MVM in video object segmentation tasks. We also observe significant performance gains across *all* tasks with models trained with smaller patch sizes (ViT-S/8, ViT-B/8). SiamMAE achieves competitive results without relying on data augmentation [16, 17], handcrafted tracking-based pretext tasks [15, 14], multi-crop training [17], additional techniques to prevent representational collapse [10, 11, 17, 16] or enhance performance [11]. We believe that our detailed analysis, straightforward approach, and state-of-the-art performance can serve as a robust baseline for self-supervised correspondence learning.

## 2  Related Work

**Temporal correspondence.** The visual world is smooth and continuous [33, 34], providing a rich source of information for biological and machine vision systems. In biological vision, infants learn about objects and their properties by establishing temporal correspondence, taking advantage of the inherent smoothness in the visual input [35]. Similarly, in machine vision, learning fine-grained correspondence from video frames is an important problem that has been studied for decades in the form of optical flow and motion estimation [36–42, 5, 6, 43, 44, 7]. However, despite their impressive performance, these methods rely on costly human-annotated or synthetic data with pixel-level ground truth annotations [45, 46]. A more semantically meaningful task involves determining object-level correspondence i.e., visual object tracking [47–52]. One popular approach is tracking-by-matching methods that utilize deep features learned via supervised [53, 54] or self-supervised learning [10–12, 14–16] on videos. State-of-the-art methods [14–17] for self-supervised feature learning for correspondence primarily employ some form of contrastive learning [13]. Predictive learning has also been used for learning correspondences [10, 11] by predicting the target colors for gray-scale input frame by observing a colorful reference frame. However, the performance of these methods has trailed behind contrastive approaches. In this work, we show that predictive learning based methods can be used for learning fine-grained and object-level correspondence.

**Self-supervised visual representation learning.** Self-supervised learning is a way to learn generalizable visual representations often using different pretext tasks for pre-training [55–59]. The presence of temporal information in videos has been leveraged in numerous ways for representation learning, including future prediction [60–62, 62–64, 10, 65], temporal ordering [66–70], object motion [71, 72, 58, 14], and temporal coherence [73, 74]. Recently, the community has made great progress in self-supervised representation learning via contrastive learning [75, 13]. Contrastive methods in the image domain encourage models to learn representations by modeling image similarity and dissimilarity [76, 19, 17, 20] or only similarity [77, 21]. Furthermore, contrastive learning has also been successfully applied to videos [78–82, 82, 83]. However, a limitation of contrastive approaches is their dependence on careful selection of augmentations to learn useful invariances [18], and as well as the need for a suite of additional components [19, 20, 17, 21] to prevent representational collapse.

**Masked autoencoders.** Masked autoencoders are a type of denoising autoencoder [84] that learn representations by reconstructing the original input from corrupted (i.e., masked) inputs. The introduction of masked language modeling in BERT [85] has had a transformative impact on the natural language processing field, particularly when scaled to large datasets and model sizes [86, 87]. Masked autoencoders have also been successfully adapted to learn representations from images [24–26] and videos [28, 27]. Our work studies a simple extension of MAEs [24] to videos. However, unlike prior methods [28, 27] that symmetrically mask all frames, we propose an asymmetric masking scheme, leaving the past frame unchanged and masking a higher percentage of the future frame.

**Siamese networks.** Siamese networks [88] are weight-sharing neural networks used to compare entities. They have been widely used in a variety of application domains [88–90] including tracking [53]. Siamese networks have been extensively used in modern contrastive learning approaches [76, 19, 17, 20], as their design allows an easy way to learn invariant visual representations from data. Inspired by the success of masked autoencoders, researchers have also explored combining contrastive learning with siamese networks and masked visual modeling [91, 92]. However, we are not aware of any previous studies that have investigated siamese masked autoencoders using asymmetric masking for representation learning from videos.

## 3  Method

Our goal is to develop a self-supervised method for learning correspondence. To that end, we study a simple extension of MAE [24] to video data (Fig. 1). In this section, we describe the key components of our Siamese Masked Autoencoder.

**Patchify.** Given a video clip with $L$ frames, we first randomly sample 2 frames $f_1$ and $f_2$. The distance between these two frames is determined by selecting a random value from the predetermined range of potential frame gaps. Following the original ViT [31], we "patchify" each frame by converting it into a sequence of non-overlapping $N \times N$ patches. Finally, position embeddings [94]

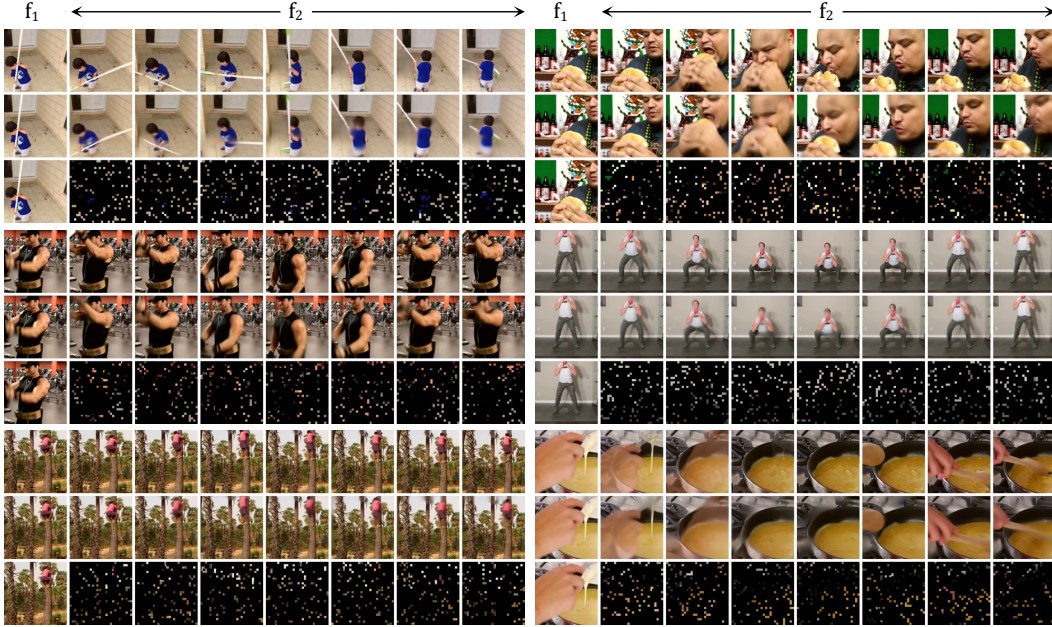

Figure 2: **Visualizations** on the Kinetics-400 [93] validation set (masking ratio 90%). For each video sequence, we sample a clip of 8 frames with a frame gap of 4 and show the original video (top), SiamMAE output (middle), and masked future frames (bottom). Reconstructions are shown with $f_1$ as the first frame of the video clip and $f_2$ as the remaining frames, using a SiamMAE pre-trained ViT-S/8 encoder with a masking ratio of 95%.

are added to the linear projections [31] of the patches, and a `[CLS]` token is appended. We do not use any temporal position embeddings.

**Masking.** Natural signals like images and videos are highly redundant, exhibiting spatial and spatio-temporal redundancies, respectively [33, 34]. To create a challenging predictive self-supervised learning task, MAEs randomly mask a high percentage (75%) of image patches [24] and extensions to videos [28, 27] use an even higher masking ratio (90%). In both images and videos, the masking strategy is symmetric, i.e., all frames have a similar masking ratio. This deliberate design choice prevents the network from leveraging and learning temporal correspondence, leading to sub-optimal performance on correspondence learning benchmarks.

We posit that *asymmetric* masking can create a challenging self-supervised learning task while encouraging the network to learn temporal correlations. Specifically, we do not mask any patches in $f_1$ (0%) and mask a very high ratio (95%) of patches in $f_2$. By providing the entire past frame as input, the network only needs to propagate the patches from the past frames to their appropriate locations in the future frame. This, in turn, encourages the network to model object motion and focus on object boundaries (Fig. 5). To further increase the difficulty of the task, we sample the two frames with a large temporal gap. Although predicting further into the future is inherently ambiguous and may yield multiple plausible outcomes, providing a small number of patches as input for the second frame results in a challenging yet tractable self-supervised learning task.

**Encoder.** We explore two different encoder configurations for processing input frames.

A *joint encoder* is a natural extension of image MAEs to a pair of frames. The unmasked patches from the two frames are concatenated and then processed by a standard ViT encoder.

A *siamese encoder* [88] are weight-sharing neural networks used for comparing entities and are an essential component of modern contrastive representation learning methods [21]. Siamese networks have been used for correspondence learning [53, 11, 10] and often require some *information bottleneck* to prevent the network from learning trivial solutions. For example, Lai and Xie [11] propose to use color channel dropout to force the network to avoid relying on colors for matching correspondences.

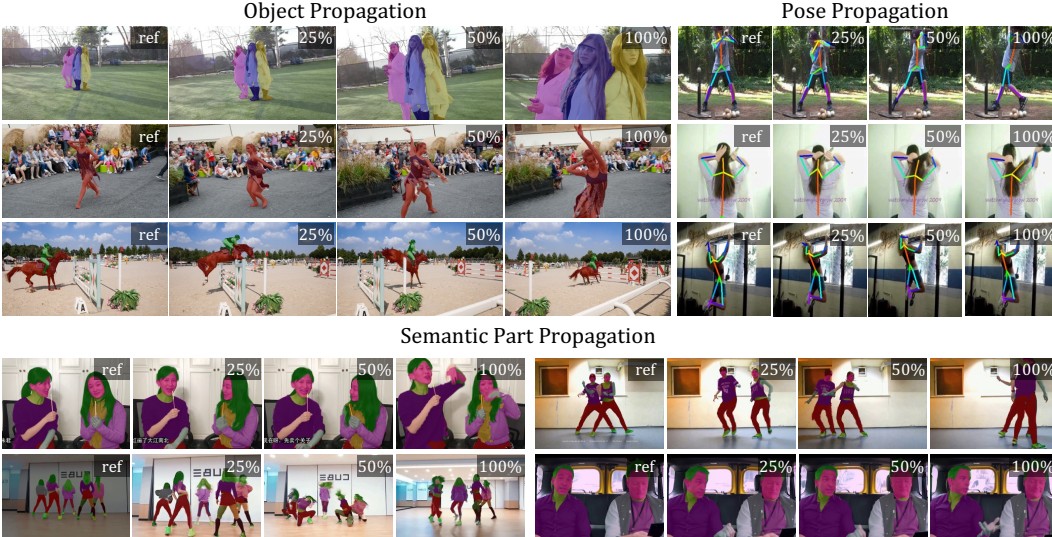

Figure 3: **Qualitative** results on three downstream tasks: video object segmentation (DAVIS-2017 [95]), human pose propagation (JHMDB [96]) and semantic part propagation (VIP [97]).

We use siamese encoders to process the two frames independently and our asymmetric masking serves as an information bottleneck.

**Decoder.** The output from the encoder is projected using a linear layer and [MASK] tokens with position embeddings are added to generate the full set of tokens corresponding to the input frame. We explore three different decoder configurations which operate on the full set of tokens.

A *joint decoder* applies vanilla Transformer blocks on the concatenation of full set of tokens from both frames. A key downside of this approach is a substantial increase in GPU memory requirement, especially when using smaller patch sizes.

A *cross-self decoder* is similar to the original encoder-decoder design of the Transformer [94] model. Each decoder block consists of a *cross-attention* layer and a *self-attention* layer. The tokens from $f_2$ attend to the tokens from $f_1$ via the cross-attention layer and then attend to each other via the self-attention layer. We note that the cross-attention layer is functionally similar to the affinity matrix often used in self-supervised correspondence learning approaches [11, 10].

A *cross decoder* consists of decoder blocks with only *cross-attention* layers, where tokens from $f_2$ attend to the tokens from $f_1$.

Finally, the output sequence of the decoder is used to predict the normalized pixel values [24] in the masked patches. $l2$ loss is applied between the prediction of the decoder and the ground truth.

## 4 Experiments

In this section, we evaluate our method on three different tasks, compare its performance with prior state-of-the-art methods, and perform extensive ablation studies of different design choices. For qualitative results, see Fig. 2, Fig. 3, Fig. 5 and videos on our project website.

### 4.1 Experimental Setup

**Backbone.** We use the ViT-S/16 model for most of our experiments as it is similar to ResNet-50 in terms of the number of parameters (21M vs 23M) and allows for fair comparisons across different self-supervised learning and correspondence learning methods.

**Pre-training.** Models are pre-trained using Kinetics-400 [93] for self-supervised learning. SiamMAE takes as input pairs of randomly sampled frames ($224 \times 224$) with a frame gap ranging from 4 to 48 frames at a rate of 30 fps. We perform *minimal* data augmentation: random resized cropping

| Method | Backbone | Dataset | DAVIS | | | VIP | JHMDB | |
|---|---|---|---|---|---|---|---|---|
| | | | $\mathcal{J}\&\mathcal{F}_m$ | $\mathcal{J}_m$ | $\mathcal{F}_m$ | mIoU | PCK@0.1 | PCK@0.2 |
| Supervised [98] | ResNet-50 | ImageNet | 66.0 | 63.7 | 68.4 | 39.5 | 59.2 | 78.3 |
| SimSiam [20] | ResNet-50 | ImageNet | 66.3 | 64.5 | 68.2 | 35.0 | 58.4 | 77.5 |
| MoCo [19] | ResNet-50 | ImageNet | 65.4 | 63.2 | 67.6 | 36.1 | 60.4 | 79.3 |
| TimeCycle [14] | ResNet-50 | VLOG | 40.7 | 41.9 | 39.4 | 28.9 | 57.7 | 78.5 |
| UVC [12] | ResNet-50 | Kinetics | 56.3 | 54.5 | 58.1 | 34.2 | 56.0 | 76.6 |
| VFS [16] | ResNet-50 | Kinetics | 68.9 | 66.5 | 71.3 | 43.2 | 60.9 | 80.7 |
| MAE-ST [27] | ViT-L/16 | Kinetics | 54.6 | 55.5 | 53.6 | 33.2 | 44.4 | 72.5 |
| MAE [24] | VIT-B/16 | ImageNet | 53.5 | 52.1 | 55.0 | 28.1 | 44.6 | 73.4 |
| VideoMAE [28] | ViT-S/16 | Kinetics | 39.3 | 39.7 | 38.9 | 23.3 | 41.0 | 67.9 |
| Dino [17] | ViT-S/16 | ImageNet | 61.8 | 60.2 | 63.4 | 36.2 | 45.6 | 75.0 |
| **SiamMAE** (ours) | ViT-S/16 | Kinetics | 62.0 | 60.3 | 63.7 | 37.3 | 47.0 | 76.1 |
| **SiamMAE** (ours) | ViT-B/16 | Kinetics | **62.8** | **60.9** | **64.6** | **38.4** | **47.2** | **76.4** |
| Dino [17] | ViT-S/8 | ImageNet | 69.9 | 66.6 | 73.1 | 39.5 | 56.5 | 80.3 |
| **SiamMAE** (ours) | ViT-S/8 | Kinetics | 71.4 | 68.4 | 74.5 | 45.9 | 61.9 | 83.8 |
| **SiamMAE** (ours) | ViT-B/8 | Kinetics | **72.3** | **68.9** | **75.6** | **46.5** | **62.4** | **84.0** |

Table 1: **Comparison with prior work** on three downstream tasks: video object segmentation (DAVIS-2017 [95]), human pose propagation (JHMDB [96]) and semantic part propagation (VIP [97]).

and horizontal flipping. Training is done for 400 epochs for the ablation studies (Table 2, 3) and for 2000 epochs for the results in Table 1. We adopt repeated sampling factor [99, 27] of 2 and report "effective epochs", i.e., the number of times a training video is viewed during training. We use the AdamW optimizer [100] with a batch size of 2048. Additional training details are in § A.

**Evaluation methodology.** We evaluate the quality of learned representations for dense correspondence task using $k$-nearest neighbor inference on three downstream tasks: video object segmentation (DAVIS-2017 [95]), human pose propagation (JHMDB [96]) and semantic part propagation (VIP [97]). Following prior work [14–16], all tasks are formulated as video label propagation: given the ground-truth label for the initial frame, the goal is to predict the label for each pixel in future frames of a video. We also provide temporal context during inference by maintaining a queue of the last $m$ frames, and we limit the set of source patches considered to a spatial neighborhood of the query patch. See § A for evaluation hyperparameters.

## 4.2 Comparison with Prior Work

**Video Object Segmentation.** We first evaluate our model on DAVIS 2017 [95], a benchmark for video object segmentation, for the task of semi-supervised multi-object segmentation. We follow the evaluation protocol of prior work and use images of a 480p resolution for evaluation. We find that SiamMAE *significantly* outperforms VideoMAE (39.3% to 62.0%), which we attribute to the use of tube masking scheme in VideoMAE which prevents the model from learning temporal correspondences. Like DINO [17], we also find that reducing the patch size leads to significant performance gains. Our ViT-S/8 (+9.4%) model outperforms *all* prior contrastive learning and self-supervised correspondence learning approaches. Finally, we note that although the larger MAE-ST models (ViT-L/16, 304M parameters) trained with random masking perform better than VideoMAE, their performance still lags SiamMAE by a considerable margin. Surprisingly, we find that MAEs trained on videos perform similarly to image MAEs. Unlike images, which are (approximately) isotropic [29], the temporal dimension is special [30], and not all spatio-temporal orientations are equally likely. Hence, symmetrically treating spatial and temporal information might be sub-optimal.

**Video Part Segmentation.** Next, we evaluate SiamMAE on the Video Instance Parsing (VIP) [97] benchmark, which involves propagating semantic masks for 20 different human parts. Compared to other datasets in our evaluation, VIP is especially challenging, as it involves much longer videos (up to 120 seconds). We follow the evaluation protocol of prior work [12], using $560 \times 560$ images and a single context frame. On this challenging task, our ViT-S/8 model substantially outperforms DINO (39.5 to 45.9). SiamMAE benefits more from smaller patch sizes than DINO, achieving an +8.6

| encoder | decoder | mask ratio | $\mathcal{J}\&\mathcal{F}_m$ | $\mathcal{J}_m$ | $\mathcal{F}_m$ |
|---------|---------|------------|------|------|------|
| joint | joint | 0.50 (s) | 51.8 | 50.7 | 52.9 |
| joint | joint | 0.75 (s) | 55.4 | 54.3 | 56.6 |
| joint | joint | 0.90 (s) | 51.9 | 50.8 | 52.9 |
| siam | cross-self | 0.95 (a) | **58.1** | **56.6** | **59.6** |

(a) **FrameMAE.** Simple extension of MAEs to frames does not work.

| encoder | decoder | $\mathcal{J}\&\mathcal{F}_m$ | $\mathcal{J}_m$ | $\mathcal{F}_m$ |
|---------|---------|------|------|------|
| joint | joint | 49.7 | 48.0 | 51.5 |
| joint | cross | 44.6 | 43.6 | 45.7 |
| joint | cross-self | 41.1 | 39.6 | 42.7 |
| siam | joint | 56.7 | 55.4 | 58.1 |
| siam | cross | 52.2 | 51.2 | 53.1 |
| siam | cross-self | **58.1** | **56.6** | **59.6** |

(b) **Encoder-decoder design.** The combination of a siamese encoder and a cross-self decoder works the best.

| mask ratio | pattern | $\mathcal{J}\&\mathcal{F}_m$ | $\mathcal{J}_m$ | $\mathcal{F}_m$ |
|------------|---------|------|------|------|
| 0.50 (s) | random | 41.5 | 40.2 | 42.7 |
| 0.50 (s) | grid | 48.2 | 46.7 | 49.7 |
| 0.75 (s) | random | 52.7 | 51.3 | 54.1 |
| 0.90 (s) | random | 51.4 | 50.0 | 52.8 |
| 0.95 (a) | random | **58.1** | **56.6** | **59.6** |

(c) **Symmetric masking.** Symmetric random masking degrades performance.

| mask ratio | $\mathcal{J}\&\mathcal{F}_m$ | $\mathcal{J}_m$ | $\mathcal{F}_m$ |
|------------|------|------|------|
| 0.50 (a) | 49.0 | 48.4 | 49.6 |
| 0.75 (a) | 55.3 | 54.1 | 56.4 |
| 0.90 (a) | 58.4 | 57.0 | 59.8 |
| 0.95 (a) | **58.1** | **56.6** | **59.6** |

(d) **Asymmetric masking.** Extremely high asymmetric masking is essential.

Table 2: **SiamMAE ablation experiments** on DAVIS [95] with the default setting: siamese encoder, cross-self decoder, asymmetric (a) masking ratio (95%), and a frame sampling gap of $[4 - 48]$. Default settings are marked in blue and (s) denotes symmetric masking.

mIoU improvement over DINO's +3.3 mIoU. Finally, SiamMAE outperforms *all* prior contrastive learning and self-supervised correspondence learning approaches.

**Pose Tracking.** We evaluate SiamMAE on the task of keypoint propagation, which involves propagating 15 keypoints and requires spatially precise correspondence. We follow the evaluation protocol of prior work [12], using $320 \times 320$ images and a single context frame. SiamMAE outperforms all prior work and benefits more from smaller patch sizes than DINO (+14.9 to +10.9 PCK@0.1).

Finally, we test the scalability of SiamMAE by training and evaluating ViT-B models. Across all three tasks, ViT-B models outperformed ViT-S models for both patch sizes tested.

## 4.3 Ablation Studies

We ablate SiamMAE to understand the contribution of each design decision with the default settings: siamese encoder, cross-self decoder, asymmetric masking ratio (95%), frame sampling gap $4 - 48$.

**FrameMAE.** We compare SiamMAE with FrameMAE (Table 2a), an extension of MAEs to video frames, i.e., *joint* encoder and *joint* decoder with symmetric masking ratio. FrameMAE performs significantly worse when the masking ratio is too high (90%) or too low (50%). With a 90% masking ratio, the task becomes challenging (higher loss) due to the insufficient number of patches available to learn temporal correspondence. When the masking ratio is 50%, the task becomes easier (lower loss) and the network can reconstruct the frames without relying on temporal information, due to the spatial redundancy of images. SiamMAE with an asymmetric masking ratio works best.

**Encoder-decoder design.** An important design decision of SiamMAE is the choice of encoder and decoder. We study the performance of various combinations of encoders and decoders with asymmetric masking in Table 2b. Joint encoders perform significantly worse compared to their siamese counterparts across all decoder designs. This can be attributed to the difference between the training and testing setups, as each frame is processed independently during the testing phase.

Siamese encoder with cross decoder performs worst among siamese encoders. We also observe that the training loss is higher and the reconstructed frames are spatially incoherent, as all patches from $f_2$ are processed independently. Finally, the combination of a siamese encoder with a cross-self decoder outperforms all other pairings. The cross-attention operation is similar to the affinity matrix used

| spatial | color | $\mathcal{J}\&\mathcal{F}_m$ | $\mathcal{J}_m$ | $\mathcal{F}_m$ |
|---|---|---|---|---|
| | | 56.8 | 55.5 | 58.1 |
| ✓ | | **58.1** | **56.6** | **59.6** |
| | ✓ | 55.8 | 54.6 | 57.0 |
| ✓ | ✓ | 56.7 | 55.4 | 57.9 |

(a) **Data augmentation.** SiamMAE requires minimal data augmentation.

| frame gap | $\mathcal{J}\&\mathcal{F}_m$ | $\mathcal{J}_m$ | $\mathcal{F}_m$ |
|---|---|---|---|
| 4 | 55.1 | 53.5 | 56.7 |
| 8 | 56.4 | 54.9 | 57.8 |
| 16 | 58.0 | 56.7 | 59.4 |
| 32 | 57.7 | 56.3 | 59.1 |
| 4-48 | **58.1** | **56.6** | **59.6** |

(b) **Frame sampling.** Random frame gap works the best.

Table 3: **Data augmentation.** We ablate the importance of manual (spatial and color jitter) and natural data augmentation (frame sampling) for learning correspondence via our SiamMAE on DAVIS [95]. The table format follows Table 2.

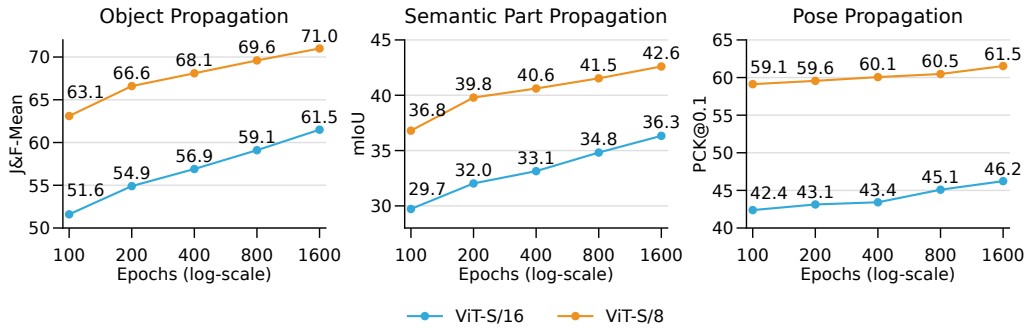

Figure 4: **Training schedule and patch size.** Evaluation of SiamMAE performance for 3 downstream tasks for ViT-S/16 and ViT-S/8 models. Across all tasks, longer training and smaller patch sizes lead to improved performance.

in self-supervised correspondence learning and is also used for label propagation in our evaluation protocol. Hence, by processing the frames independently and decoding them via the cross-self decoder, the network is encouraged to learn good representations for dense visual correspondence.

**Masking.** Next, we discuss the effect of the masking scheme for the combination of a siamese encoder with a self-cross-decoder. Random symmetric masking performs poorly and is also worse than the corresponding FrameMAE configurations (Table 2a, 2c). We also study the *grid-wise* mask sampling strategy, which keeps every alternate patch. This is an easier task, as the masking pattern enables the network to exploit and learn spatio-temporal correlations. Although we see significant gains (41.5 to 48.2), performance is still significantly poor compared to SiamMAE. In Table 2d, we study the role of different asymmetric masking ratios. We notice a clear trend: increasing the masking ratio from $50\%$ to $95\%$ increases the performance ($49.0\%$ to $58.1\%$).

**Data augmentation.** In Table 3a we study the influence of different data augmentation strategies. Similar to the findings in the image [24] and video [27] domains, we find that SiamMAE does not require extensive data augmentation to achieve competitive performance. Random cropping with a scale range of $[0.5, 1]$ and horizontal flipping works best, and adding color jitter leads to performance degradation. Contrastive methods like DINO show impressive $k$-NN performance by using extensive data augmentation. In contrast SiamMAE achieves superior results by relying on natural data augmentation available in videos, discussed next.

**Frame sampling.** Video data is a rich source of data augmentation, e.g. variations in pose, lighting viewpoint, occlusions, etc. To effectively leverage this, we study the importance of frame sampling in Table 3b. The performance improves as we increase the frame sampling gap. Natural videos frequently exhibit gradual temporal changes; therefore, increasing the frame interval results in a more robust natural data augmentation, which in turn enhances performance. Our frame sampling strategy is simple and effective: randomly sample frames with a frame gap ranging from 4 to 48 frames.

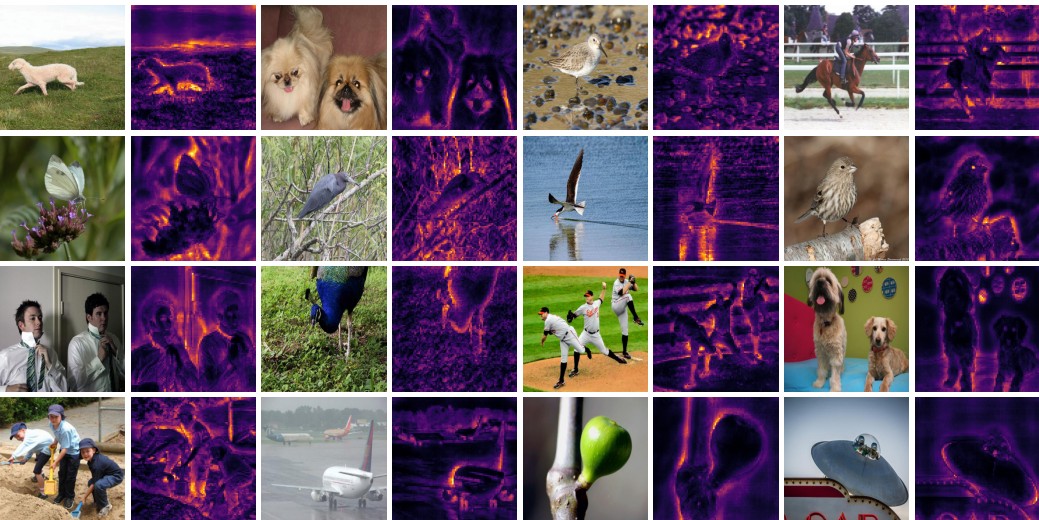

Figure 5: **Self-attention maps.** Self-attention maps from a ViT-S/8 model. We examine the self-attention of the [CLS] token on the heads of the final layer. Unlike contrastive methods, there is no explicit loss function acting on the [CLS] token. These self-attention maps suggest that the model has learned the notion of object boundaries from object motion in videos. See project page for videos.

**Training schedule.** As noted earlier, our ablations are based on 400-epoch pre-training. Figure 4 studies the influence of the training length schedule for ViT-S/16 and ViT-S/8 models for the three downstream tasks considered in this work. Due to compute limitations, we report evaluations of a single model at different checkpoints. Across both patch sizes and across all tasks, the accuracy improves gradually with longer training.

**Prediction target.** In Table 5a (see § B) we study the importance of predicting the future. We consider two additional SiamMAE variations: one where we always predict the past frame (f1) and another where the order of frame prediction (f1 or f2) is randomized. All variations perform reasonably well, with our default setting (i.e., predicting the future) performing the best. We emphasize predicting future behavior due to its natural alignment with most real-world applications, which often necessitate the anticipation or prediction of agents' future behavior.

### 4.4 Attention Map Analysis

In Fig. 5, we visualize the self-attention map of the ViT-S/8 model. We use the [CLS] token as the query and visualize the attention of a single head from the last layer with 720p images from ImageNet. We find that the model attends to the object boundaries. For instance, it can clearly delineate iconic objects (such as the sheep in the first row, first column), multiple objects (like the three baseball players in the third row, sixth column), and even when the scene is cluttered (as seen with the bird in the second row, fourth column). While other self-supervised learning approaches [17, 101] have reported emergent object segmentation capabilities, we are unaware of any methods demonstrating an *emergent* ability to predict object boundaries. This *emergent* ability is unique and surprising since, unlike contrastive learning approaches, no loss function operates on the [CLS] token in SiamMAE (or in MAEs). We attribute the emergence of this ability to our asymmetric masking ratio, which encourages the model to learn about object boundaries from object motion in videos.

### 4.5 Failure Analysis

We evaluate the quality of learnt representations using label propagation and consequently inherit its limitations. Specifically, the inference algorithm lacks semantic understanding, leading to globally inconsistent labels (Fig 6). This limitation can be overcome by fine tuning the learnt representations with task specific architectural changes. Additionally, there are instances where the inference process

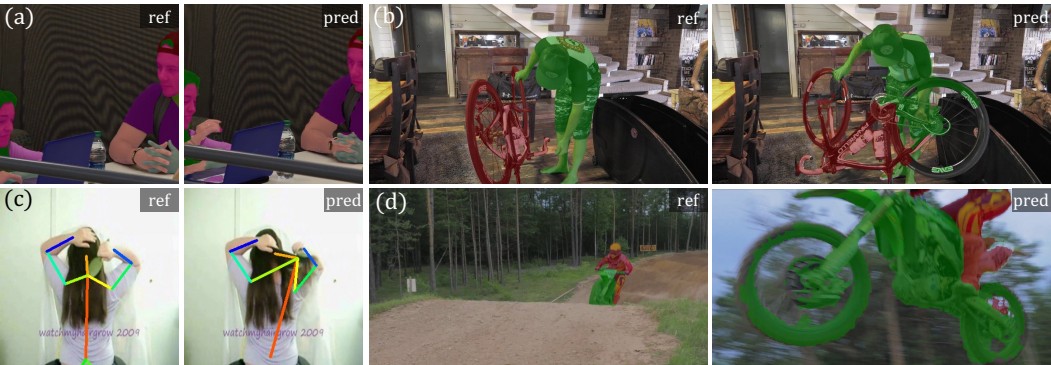

Figure 6: **Failure analysis.** A key disadvantage of using label propagation is the lack of global semantic understanding of objects. Assigning labels based solely on low-level features can lead to globally inconsistent labels, as illustrated by the following examples: (a) a segmentation mask that covers both hands; (b) a pose key-point determined using the person's hair, rather than their posture; (c) challenges in assigning labels to parts of the object that are occluded in the reference frame; and (d) the inability to assign labels to fine object details, such as the spokes of a tire.

might miss intricate object details, like the spokes of a tire. While this shortcoming can be mitigated by using a smaller patch size during training and inference, it comes at a higher compute cost.

## 5    Conclusion

In this work, we introduce SiamMAE, a simple method for representation learning from videos. Our approach is based on the intuition that the temporal dimension should be treated differently from the spatial dimension. We demonstrate that an *asymmetric* masking strategy, i.e., masking a high percentage of patches of the future frame while keeping the past frame unchanged, is an effective strategy for learning correspondence. By predicting a majority fraction of the future frame, we find that our SiamMAE is able to learn the notion of object boundaries (Fig 5). Moreover, unlike MAEs, features learned via our approach can be used in a zero-shot manner and outperform state-of-the-art self-supervised methods in various tasks, such as video object segmentation, pose keypoint propagation, and semantic part propagation. SiamMAE achieves these competitive results without the need for data augmentation, handcrafted tracking-based pretext tasks, or other techniques to prevent representational collapse. We hope our work will encourage further exploration of learning representations by predicting the future.

**Future work.** Our study focuses on learning correspondences by operating on pairs of video frames. This choice was driven by the empirical success of the approach and the limited computational resources available. Consequently, we believe that further investigation is needed to understand the role of predicting multiple future frames based on past frames, both for general visual representation learning and for correspondence learning specifically. An important future direction is to systematically examine the scalability of our approach in terms of both data and model size. Following previous work, we utilize internet videos for pre-training. However, it is essential to also investigate the impact of different types of video data, such as egocentric videos [102] versus "*in-the-wild*" internet videos. Lastly, our learned representations hold potential for applications involving embodied agents (i.e., robots), as the concept of correspondence could be useful in tasks such as object manipulation, navigation, and interaction within dynamic environments.

**Acknowledgments.**    We thank Abhishek Kadian for helpful discussions. This research was in part supported by the Stanford Institute for Human-Centered Artificial Intelligence (HAI) and ONR MURI N00014-22-1-2740.

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

## A  Implementation Details

**Training.** Our training settings follow [24] and we build on the open-source implementation of MAEs (`https://github.com/facebookresearch/mae`) for all our experiments. We use the parameters specified in the original implementation unless specified otherwise in Table 4a. All our experiments are performed on 4 Nvidia Titan RTX GPUs for ViT-S/16 models, and on 8 Nvidia Titan RTX GPUs for ViT-S/8 models and ViT-B models.

**Evaluation methodology.** Our evaluation methodology follows prior work [14–16] and in Table 1 we report results previously reported in these studies. For recent self-supervised learning approaches like DINO, MAEs, MAE-ST and VideoMAE, we carry out a comprehensive grid search on the evaluation hyperparameters listed in Table 4b, and report the optimal results obtained. The evaluation parameters for SiamMAE can be found in Table 4b.

| config | value |
|---|---|
| optimizer | AdamW [100] |
| optimizer momentum | $\beta_1, \beta_2{=}0.9, 0.95$ [103] |
| weight decay | 0.05 |
| learning rate | 1.5e-4 |
| learning rate schedule | cosine decay [104] |
| warmup epochs [105] | 40 |
| epochs | 2000 (ablations 400) |
| repeated sampling [99] | 2 |
| augmentation | hflip, crop $[0.5, 1]$ |
| batch size | 2048 (S) 1024 (B) |
| frame sampling gap | $[4, 48]$ |

| config | DAVIS | VIP | JHMDB |
|---|---|---|---|
| top-k | 7 | 10 | 7 |
| queue length | 20 | 20 | 20 |
| neighborhood size | 20 | 8 | 20 |

(a) **Kinetics pre-training setting.**  (b) **Evaluation setting.**

Table 4: Training and evaluation hyperparameters.

## B  Additional Ablations

**Prediction target.** In Table 5a we study the importance of predicting the future. We consider two additional SiamMAE variations: one where we always predict the past frame (f1) and another where the order of frame prediction (f1 or f2) is randomized. All variations perform reasonably well, with our default setting (i.e., predicting the future) performing the best. We emphasize predicting future behavior due to its natural alignment with most real-world applications, which often necessitate the anticipation or prediction of agents' future behavior.

**Frame overlap analysis.** To perform frame overlap analysis, we sampled video frames from the Kinetics-400 validation set with the specified frame gap and calculated two image similarity metrics: mean squared error (mse) and structural similarity index measure (ssim). We observed that either a very high overlap (low frame gap, high ssim, and low mse) or a low overlap (high frame gap, low ssim, and high mse) adversely affects performance. Sampling with a frame gap of 16 or within a range of $[4, 48]$ yields the best results. Interestingly, the overlap metrics for a frame gap of 16 and $[4, 48]$ are comparable, suggesting that a particular degree of overlap is important for best results.

| pred. target | $\mathcal{J}\&\mathcal{F}_m$ | $\mathcal{J}_m$ | $\mathcal{F}_m$ |
|---|---|---|---|
| f1 (past) | 57.5 | 56.0 | 59.0 |
| random (f1, f2) | 57.8 | 56.3 | 59.2 |
| f2 (future) | **58.1** | **56.6** | **59.6** |

| frame gap | ssim | mse | $\mathcal{J}\&\mathcal{F}_m$ |
|---|---|---|---|
| 4 | 0.6231 | 0.0230 | 55.1 |
| 8 | 0.5343 | 0.0360 | 56.4 |
| 16 | 0.4749 | 0.0480 | 58.0 |
| 32 | 0.4221 | 0.0597 | 57.7 |
| 4-48 | 0.4548 | 0.0528 | **58.1** |

(a) **Prediction target.** Predicting the future works the best.

(b) **Frame sampling.** Random frame gap works the best.

Table 5: **Additional ablations.** We ablate the importance of predicting the future and perform overlap analysis for different frame gaps. The table format follows Table 2.

