# OpenReview forum: "Siamese Masked Autoencoders"
_NeurIPS.cc/2023/Conference — NeurIPS 2023 oral_

### Official Review · Reviewer_5RkZ · 2023-06-27

**Soundness:** 4 excellent
**Presentation:** 4 excellent
**Contribution:** 3 good
**Rating:** 7
**Confidence:** 4

**Summary:**

In this paper, the authors propose a simple extension to Masked Autoencoders (MAE) to be able to pre-train on videos: SiamMAE. Two frames are sampled, independently encoded, and then asymmetrically masked. A transformer decoder is used to predict the missing patches in the masked image. The authors show that by masking a high proportion of patches (0.95) in the future frame and leaving the past frame unmasked, they are able to encourage the network to learn a more object-centric representation and focus on object motion rather than low-level image details. The authors show that this simple approach outperforms previous methods on many down-stream tasks and perform an extensive ablation to examine the architecture choices.

**Strengths:**

Very well written paper. Some results are particularly impressive (e.g. more than 20% gain over VideoMAE)

Simple but effective method and the honesty of presenting this as-is (instead of disguising the method as being more complicated than it needs to be) I think should be appreciated.

Good comparison to other work and a great ablation section explaining many design choices


**Weaknesses:**

Can only see two minor weaknesses:

1.	The approach adds little over MAE and feels incremental, however it works very well and hasn’t been done before.
2.	This approach relies on the temporal smoothness that is found in many curated datasets, such as Kinetics. However, for in-the-wild videos, with many sharp scene changes, this assumption becomes less likely to hold. However, it should be possible to split long videos into scenes in an unsupervised manner and then sample frames within a scene. It would be nice to a discussion on how this can be applied to large-scale internet datasets (which are becoming very popular for foundation models)


**Questions:**

With standard MAE, it is possible to also mask the input images for the downstream tasks to speed-up computation. It seems that this avenue for speed-up would not be possible with this current approach since the previous frame has to be unmasked?

Could the authors clarify the reason for not including temporal positional encoding (was this ablated?)

It would be interesting to see a bit more how this frame-based method compares to video-based methods on downstream tasks (for example more comparisons like VideoMAE). Perhaps focusing on compute efficiency?


**Limitations:**

Yes

---

> ### Author Rebuttal · Authors · 2023-08-09
>
> Thank you for your comments and suggestions. We address the reviewer concerns below:
>
> >*Novelty over MAE*
>
> We agree that our method is a simple modification of MAE albeit one which has not been explored in the past.  We hope that the simplicity, efficacy and extensive empirical analysis of our method is a valuable contribution to the community.
>
> >*Application to “in-the-wild” datasets*
>
> Predicting the future frame based on the past frame to learn correspondence is effective because the two frames often have some degree of overlap. We consider a maximum frame gap of 48 frames, equivalent to 1.6s at 30fps. Over such short time horizons, temporal smoothness is a reasonable assumption. However, we agree that "in-the-wild" datasets, particularly egocentric datasets like Ego4D, might have frequent sharp scene changes due to head movements. Exploring these datasets is a promising future direction (refer to L298), especially when examining if the current strategy of random sampling remains effective. We will include this discussion in the revised text.
>
> >*Masking input images for downstream tasks to speed-up computation*
>
> To the best of our knowledge the masking strategy of MAEs is known to speed up the training process as the encoder acts on a small set of unmasked tokens. During inference generally the entire image is processed by the encoder [1, 2]. It is possible that some works have done inference on masked images however we are not aware of such works.
>
> [1] He, Kaiming, et al. "Masked autoencoders are scalable vision learners." Proceedings of the IEEE/CVF conference on computer vision and pattern recognition. 2022.
>
> [2] Li, Yanghao, et al. "Scaling language-image pre-training via masking." Proceedings of the IEEE/CVF Conference on Computer Vision and Pattern Recognition. 2023.
>
> >*Reason for no temporal position encoding*
>
> Temporal position embeddings (TPE) enable the network in distinguishing between tokens from distinct frames. In SiamMAE, the encoder processes each frame individually and the sole interaction between tokens of different frames occurs through the cross-attention layer in our cross-self decoder. Hence, the network does not need TPEs for achieving good performance. We validated this hypothesis by conducting the proposed ablation study, adding TPEs to the decoder. When using a joint decoder, TPEs improve the performance. Ideally, if position embeddings are redundant, the network should achieve similar performance with zero embedding weights. However, in practice we found that when using a cross-self decoder, the presence of TPEs hurts the performance.
>
>
> | encoder     | decoder        | temporal pos. embed  |  J & Fm   |
> | ----------- | -------------- | :------------------: | --------- |
> | siam        | joint          | &#10003;             | 57.3      |
> | siam        | joint          |                      | 56.7      |
> | siam        | cross-self     | &#10003;             | 57.6      |
> | **siam**    | **cross-self** |                      | **58.1**  |
>
> >*Comparison on additional downstream tasks*
>
> We address this in our global response and re-iterate here for convenience.
>
> We agree with the general sentiment of the comment, emphasizing the evaluation of a self-supervised representation learning method across a wide array of tasks. While we aim to demonstrate the versatility of our method, following prior work [12, 14, 16], we've limited our claims and experiments to validate our method's effectiveness at learning representations for visual correspondence. A key advantage of current evaluation strategy is its computational efficiency as it requires no training. Evaluation on video recognition benchmarks often involves fine tuning models for 300 epochs (as seen in VideoMAE, MAE-ST), requiring approximately 32-64 industrial-grade GPUs (such as V100, A100).

---

> > ### Comment · Reviewer_5RkZ · 2023-08-18
> >
> > Dear authors, thank you for addressing the comments in my review. I feel you have adequately done so.

---

### Official Review · Reviewer_uZFR · 2023-06-27

**Soundness:** 3 good
**Presentation:** 3 good
**Contribution:** 3 good
**Rating:** 6
**Confidence:** 4

**Summary:**

The paper proposes to use Siamese Masked Encoders for establishing correspondence for video input data. Uses the concept of predictive learning based on Masked Auto Encoder. Paper proposes to use asymmetric masking for present and future frames. Achieves best results in self-supervised setting for video label propagation tasks.


**Strengths:**

Strengths:
1) The paper is well written for most of the parts and aptly elucidates the advantage of using a masked auto-encoder-based method for object-based correspondence.
2) Discusses in detail the architecture design choices for the encoder and decoder, ans uses a final design which is intuitive and simple and  also focuses on relevant ablation studies.
3) The paper shows and achieves better results for object segmentation, part segmentation, and pose propagation tasks.



**Weaknesses:**

Weakness:
1) Frame sampling: Table 3b discusses the effect of frame gap, however, it misses to comment on what could be the minimum frame overlap between the pair of frames for which predictions are out to be done. Apart from asymmetric masking, the degree of overlap between consecutive frames is also an important factor.

2) Fails to discuss the probable failure cases given the limitations from more qualitative results analysis belonging to various tasks.

**Questions:**

Questions:
1) Ablation experiment: Comparison with FrameMAE. In case of FrameMAE, When we do provide 2 frames to the joint encoder, a better comparison would be comparing the asymmetric masking with FrameMAE, since from Table 2 (b, d) results as presented does not seem too behind for FrameMAE.

3) Section 4.4: This subsection would be more aptly titled as "Futher Insights" rather Qualitative Results.

**Limitations:**

Overall the idea presented in the paper is lucid and necessary details regarding reproducing the experiments are mentioned. I think the paper puts a right step in the direction of improving well know correspondence (object level) problem using predictive learning. However I feel what this paper lacks is drawing a complete comparison for an image overlap percentage based study. Presenting zero-shot results for occluded scenes and object parts and discussing failure cases (as seen from qualitative results obtained)

---

> ### Author Rebuttal · Authors · 2023-08-09
>
> Thank you for your comments and suggestions. We address the reviewer concerns below:
>
> >*Frame sampling with overlap analysis*
>
> To perform overlap analysis, we sampled video frames from the Kinetics-400 validation set with the specified frame gap and calculated two image similarity metrics: mean squared error (mse) and structural similarity index measure (ssim). We observed that either a very high overlap (low frame gap, high ssim, and low mse) or a low overlap (high frame gap, low ssim, and high mse) adversely affects performance.
> Sampling with a frame gap of 16 or within a range of [4, 48] yields the best results. Interestingly, the overlap metrics for a frame gap of 16 and [4, 48] are comparable, suggesting that a particular degree of overlap is important for best results.
>
>
> | frame gap  | ssim     | mse     | J & Fm    |
> | ---------- | -------- | ------- | --------- |
> | 4          | 0.6231   | 0.0230  | 56.7      |
> | 8          | 0.5343   | 0.0360  | 57.8      |
> | 16         | 0.4749   | 0.0480  | 58.0      |
> | 32         | 0.4221   | 0.0597  | 56.3      |
> | 4-48       | 0.4548   | 0.0528  | 58.1      |
>
> >*Qualitative analysis & discussion of failures*
>
> We've included a qualitative failure analysis in a file attached to the global response and will incorporate it into the paper.
>
> We evaluate the quality of learnt representations using label propagation and consequently inherit its limitations. Specifically, the inference algorithm lacks semantic understanding, leading to globally inconsistent labels (refer to examples in the figure). This limitation can be overcome by fine tuning the learnt representations with task specific architectural changes. Additionally, there are instances where the inference process might miss intricate object details, like the spokes of a tire. While this shortcoming can be mitigated by using a smaller patch size during training and inference, it comes at a higher compute cost.
>
> >*Comparing asymmetric masking with FrameMAE*
>
> The results in Table 2b include the comparison suggested i.e. FrameMAE with asymmetric masking. We will update the text to clarify this. For completeness the table below compares relevant FrameMAE variations with SiamMAE. The combination of siamese encoder, a cross-self decoder with asymmetric masking works the best. In the table below (a) denotes asymmetric masking and (s) denotes symmetric masking.
>
> | encoder     | decoder        | mask ratio   |  J & Fm   |
> | ----------- | -------------- | ------------ | --------- |
> | joint       | joint          | 0.50 (s)     | 51.8      |
> | joint       | joint          | 0.75 (s)     | 55.4      |
> | joint       | joint          | 0.90 (s)     | 51.9      |
> | joint       | joint          | 0.95 (a)     | 49.7      |
> | **siam**    | **cross-self** | **0.95 (a)** | **58.1**  |
>
> >*Change title of section 4.4 to Further Insights*
>
> We will update the text to incorporate the suggestion.

---

> ### Comment · Reviewer_uZFR · 2023-08-17
> **Reply to Authors**
>
> Most of my concerns has been answered. I would expect authors to incorporate some of the changes pointed out (and accepted) by authors to be reflected in the final version of the paper. I would like to maintain my ratings.

---

### Official Review · Reviewer_h4WK · 2023-07-05

**Soundness:** 3 good
**Presentation:** 4 excellent
**Contribution:** 3 good
**Rating:** 7
**Confidence:** 5

**Summary:**

* This paper propose Siamese Masked Autoencoders for learning visual correspondence from videos called SiamMAE.

* SiamMAE randomly sample a pair of video frames and randomly mask 95% of patches of the future frame, and the pair of video frames are passed into visual encoder(VIT), and cross attention decoder to reconstruct the target.

* The authors conduct several experiments on downstream tasks(vos, human pose propagation), showing superior performance.

**Strengths:**

* The paper presents a simple, yet highly effective method for the challenging video self-supervised framework.

* The proposed method is well motivated and intuitive with excellent performance.

* The ablation is sufficient and the writing is excellent.

**Weaknesses:**

* Experiments on video recognition experiments should be reported.

* Although SiamMAE is pretraining on video frames, i still think the authors should conduct experiments on image downstream tasks(such coco detection, segmentation) to show image representation capacity.

* In Table1, The previous video ssl method(VPS) is also pretrained on Kinetics with ResNet50，however，the performance of SiamMAE with ViT-S/8 is similar in DAVIS & VIP & JHMDB, any explanation for this?

**Questions:**

please refer to the strength and weakness

**Limitations:**

no limitations

---

> ### Author Rebuttal · Authors · 2023-08-09
>
> Thank you for your comments and suggestions. We address the reviewer concerns below:
>
> >*Experiments on video and image recognition*
>
> We address this in our global response and re-iterate here for convenience.
>
> We agree with the general sentiment of the comment, emphasizing the evaluation of a self-supervised representation learning method across a wide array of tasks. While we aim to demonstrate the versatility of our method, following prior work [12, 14, 16], we've limited our claims and experiments to validate our method's effectiveness at learning representations for visual correspondence. A key advantage of current evaluation strategy is its computational efficiency as it requires no training. Evaluation on video recognition benchmarks often involves fine tuning models for 300 epochs (as seen in VideoMAE, MAE-ST), requiring approximately 32-64 industrial-grade GPUs (such as V100, A100).
>
> >*VFS performance*
>
> VFS is a state-of-the-art contrastive self-supervised representation learning method for visual correspondence. VFS learns representations by maximizing similarity across different frames from the same video. This training objective is directly aligned with how the downstream performance is measured i.e., by calculating the similarity of patches for label propagation.
>
> In this work, our goal was to develop a predictive learning method which can match or outperform contrastive learning approaches for learning visual correspondence. Historically, the performance of predictive learning methods has trailed behind contrastive self-supervised learning approaches.
> This can be attributed to the training objective of predictive learning methods like SiamMAE, which focuses on low level pixel details and is not directly aligned with the downstream evaluation procedure.
> Despite this shared disadvantage, we significantly outperform prior predictive learning based methods (+22.7 improvement over VideoMAE). Finally, we achieve our stated goal as the ViT-S/8 model improves over VFS by +2.5 J&Fm on DAVIS, +2.7 mIOU on VIP, and +1.0 PCK\@0.1 on JHMDB.

---

> > ### Comment · Reviewer_h4WK · 2023-08-20
> >
> > Thanks for the response, I have no further questions and will keep my initial rating.

---

### Official Review · Reviewer_9jyT · 2023-07-06

**Soundness:** 3 good
**Presentation:** 3 good
**Contribution:** 3 good
**Rating:** 7
**Confidence:** 3

**Summary:**

This paper focuses on the self-supervised learning for video representations. The proposed SiamMAE operates on pairs of randomly sampled video frames and asymmetrically masks them, and then predicts the missing patches for visual representation learning. SiamMAE achieves significant performance and outperforms state-of-the-art self-supervised methods on video object segmentation, pose keypoint propagation, and semantic part propagation tasks.


**Strengths:**

1 The motivation is clear and strong.

2 The proposed asymmetric masking and cross-self decoder are effective and achieve good performance.


**Weaknesses:**

1 This paper mainly evaluates the proposed method on tracking problems. How is the performance on video classification tasks, such as UCF101 and HMDB51?

2 In Table 2(c), grid mask achieves better performance than random mask with 0.5 mask ratio. Increasing the mask ratio will improve the performance of random masking. How is the performance when increasing the mask ratio for grid masking?

3 This paper only investigates the ViT-S backbone. It is better to also leverage larger models, such as ViT-B, to verify the effectiveness of the proposed method.


**Questions:**

My concerns mainly lie in the experiments.

**Limitations:**

Yes

---

> ### Author Rebuttal · Authors · 2023-08-09
>
> Thank you for your comments and suggestions. We address the reviewer concerns below:
>
> >*Results on video classification tasks*
>
> We address this in our global response and re-iterate here for convenience.
>
> We agree with the general sentiment of the comment, emphasizing the evaluation of a self-supervised representation learning method across a wide array of tasks. While we aim to demonstrate the versatility of our method, following prior work [12, 14, 16], we've limited our claims and experiments to validate our method's effectiveness at learning representations for visual correspondence. A key advantage of current evaluation strategy is its computational efficiency as it requires no training. Evaluation on video recognition benchmarks often involves fine tuning models for 300 epochs (as seen in VideoMAE, MAE-ST), requiring approximately 32-64 industrial-grade GPUs (such as V100, A100).
>
> >*Role of grid masking*
>
> As suggested, we conducted additional ablations for grid masking with different masking ratios. We find that the performance increases when we increase the grid masking ratio to 0.75. However, it decreases when we further increase the masking ratio to 0.95. An advantage of grid masking is that the masking pattern encourages the network to exploit spatio-temporal correlations. However, with a very high masking ratio the network can no longer rely on temporal correlations, leading to worse performance.
>
> All the settings for the results below follow Table 2c.
>
> | mask ratio      | pattern      | J & Fm    |
> | --------------- | ------------ | --------- |
> | 0.50 (s)        | grid         | 48.2      |
> | 0.75 (s)        | grid         | 53.8      |
> | 0.95 (s)        | grid         | 49.0      |
> |  **0.95 (a)**   | **random**   | **58.1**  |
>
> >*Results on larger backbones ViT-B*
>
> We address this in our global response and re-iterate here for convenience.
>
> We note that ViT-S has approximately the same number of parameters as ResNet-50, enabling us to compare our method across a diverse set of baselines. Moreover, ResNet-50 is the largest backbone explored by prior work [12, 14, 16], all of which aimed at improving representations for correspondence. However, we agree that a systematic investigation into the scalability of our method in terms of model size (refer to L297) would be valuable. Practically speaking, we are constrained by resources. For instance, training a ViT-S/8 model on 8 Titan RTX GPUs for 2000 epochs requires approximately 16 days. Given the compute and time constraints, we trained a ViT-B/16 model for 400 epochs.
>
> | Model      | J & Fm    |
> | ---------- | --------- |
> | ViT-S/16   | 58.1      |
> | ViT-B/16   | 58.6      |
>
> The improvement in performance is encouraging. To contextualize the magnitude of the improvement, we note that scaling DINO from ViT-S/16 to ViT-B/16 results in an improvement of 0.5 J&Fm.

---

> > ### Comment · Reviewer_9jyT · 2023-08-17
> >
> > Most of my concerns are addressed. However, I still feel it is important to demonstrate the effectiveness of the method in the video classification task. HMDB51 is a small-scale dataset with only 3.5k training clips. Evaluating the method on this dataset will not introduce too much computational cost but can further verify the results.

---

> > > ### Author Response · Authors · 2023-08-21
> > > **Response to Reviewer**
> > >
> > > Thank you for your comments and suggestions. We hope to include some results on activity recognition benchmarks in the final version.

---

### Official Review · Reviewer_e5XC · 2023-07-12

**Soundness:** 4 excellent
**Presentation:** 4 excellent
**Contribution:** 3 good
**Rating:** 7
**Confidence:** 4

**Summary:**

This paper considers the problem of using self-supervised learning from video to obtain a representation that is well-suited to the task of estimating correspondence between a pair of images. They propose a significant modification of the MAE training procedure which is adapted for estimating correspondence: one image is not masked at all while the other has most (90%+) of its tokens masked, and the same (i.e. a siamese) encoder is applied to both. This is designed to require the model to internally establish correspondence. The model is trained for pixel prediction on Kinetics-400 and evaluated on several propagation tasks (object mask, part masks, human pose) using kNN inference to establish a dense correspondence field. SiamMAE is shown to greatly outperform existing self-supervised learning procedures with comparable backbone architectures, including methods trained on video. Ablative experiments confirm the importance of combining a siamese encoder with asymmetric masking. Visualization of the attention maps show that the model pays strong attention to object boundaries, seemingly a novel attribute.

**Strengths:**

1. Good motivation and contextualization with respect to past work.
1. The modification of the MAE procedure is simple but clever, and manages to extract much more information for correspondence from video than past methods.
1. Visualization of predicted images is quite impressive (Figure 2), despite the main goal being to learn a feature extractor for correspondence.
1. Comprehensive evaluation with 3 different tasks and wide selection of relevant baselines.
1. Ablative experiments verify the importance of each component of the design.
1. Hyper-parameters provided for reproducibility.

**Weaknesses:**

1. I'm not sure about the emphasis on predicting the _future_. It seems that the temporal order could be reversed (i.e. predict the past given the future) or randomized and I would expect similar results. Has this already been tested?
1. It wasn't abundantly clear how the patch-patch similarity was obtained. It seems to be taken from the cross-attention values within the decoder (line 239). However, this could be more clear since the decoder may contain multiple cross-attention layers with each having multiple heads?
1. It wasn't clear how k-NN and the queue were used to perform propagation. This should be explained in more detail or a reference provided.
1. No code provided at this stage.
1. Lack of confidence intervals (not a major issue - delta is quite large in most cases).
1. No evaluation of ViT-B and ViT-L (not a major issue - impressive results obtained with smaller model).

**Questions:**

(Please also address or correct weaknesses above.)

1. What is the purpose of the [CLS] tokens? Is the model much less effective without it? (It is surprising that the attention masks for these tokens were so salient given that they are not involved directly in the loss.)

Suggestions:
1. I wonder whether it's possible to identify a general principle/strategy for preventing shortcut learning, of which this is just one instance?

Minor edits: (no need to respond)
1. Bold values in Tables 2d and 3b seem incorrect (e.g. 58.4 > 58.1, 56.7 > 56.5).

**Limitations:**

Additional limitation:
1. Even though Kinetics-400 was used without labels, it is possible that its image distribution is quite similar to that of the downstream tasks. It would be good to discuss this potentialityu, and say that the impact of dataset similarity has not been investigated?

I do not see any negative societal impacts stemming from this individual paper.

---

> ### Author Rebuttal · Authors · 2023-08-09
>
> Thank you for your comments and suggestions. We address the reviewer concerns below:
>
> >*Emphasis on predicting the future*
>
> We agree with the reviewer that reversing the temporal order should not significantly alter the results. We conducted two additional ablation studies: one where we always predict the past frame (f1) and another where the order of frame prediction (f1 or f2) is randomized. All the settings for the results below follow Table 2.
>
> | prediction target | J & Fm    |
> | ----------------- | --------- |
> | f1 (past)         | 57.5      |
> | random [f1, f2]   | 57.8      |
> | **f2 (future)**   | **58.1**  |
>
> All settings perform reasonably well, with our default setting (i.e., predicting the future) performing the best. We emphasize predicting future behavior due to its natural alignment with most real-world applications, which often necessitate the anticipation or prediction of agents' future behavior. We will update the text with this discussion and ablation.
>
> >*Patch-patch similarity calculation*
>
> We follow prior work on representation learning (e.g. MAE) and use the output of the encoder for calculating patch-patch similarity. The decoder is only used during pre-training. We will update the text to clarify the same.
>
> >*Evaluation methodology*
>
> Our evaluation methodology follows prior work and we provide a reference for the same in the text (please see L-189, references 14-16). For completeness we provide a short description here.
>
> All evaluation tasks are cast as video label propagation, where the goal is to predict labels for each pixel in the target frames of a video, using only the ground-truth of the initial frame (i.e., the source). We measure the cosine similarity of each pixel, or patch, in the target frame with all the patches within its spatial neighborhood from the preceding m frames. The label assignment is then based on the labels of the top-k patches that have the highest similarity. Please note, the term 'queue' in this context refers to the usage of predicted labels from the past m frames.
>
> >*Code release*
>
> We will release the code and pre-trained checkpoints upon acceptance. For reproducibility we build on the open source implementation of MAE and provide all the relevant hyper parameters in the appendix.
>
> >*Confidence intervals*
>
> Our inference procedure is fully deterministic given a pre-trained model. We confirmed the same by running our inference with 5 different seeds for our ViT-S/16 models trained for 2000 epochs.
>
> | Model    | Run 1     | Run 2     | Run 3     | Run 4     | Run 5     |
> | -------- | --------- | --------- | --------- | --------- | --------- |
> | ViT-S/16 | 62.0      | 62.0      | 62.0      | 62.0      | 62.0      |
>
> >*Larger models ViT-B & ViT-L*
>
> We address this in our global response and re-iterate here for convenience.
>
> We note that ViT-S has approximately the same number of parameters as ResNet-50, enabling us to compare our method across a diverse set of baselines. Moreover, ResNet-50 is the largest backbone explored by prior work [12, 14, 16], all of which aimed at improving representations for correspondence. However, we agree that a systematic investigation into the scalability of our method in terms of model size (refer to L297) would be valuable. Practically speaking, we are constrained by resources. For instance, training a ViT-S/8 model on 8 Titan RTX GPUs for 2000 epochs requires approximately 16 days. Given the compute and time constraints, we trained a ViT-B/16 model for 400 epochs.
>
> | Model      | J & Fm    |
> | ---------- | --------- |
> | ViT-S/16   | 58.1      |
> | ViT-B/16   | 58.6      |
>
> The improvement in performance is encouraging. To contextualize the magnitude of the improvement, we note that scaling DINO from ViT-S/16 to ViT-B/16 results in an improvement of 0.5 J&Fm.
>
> >*Role of [CLS] token*
>
> Following the original ViT paper which appends the [CLS] token during supervised training and uses the output corresponding to the [CLS] token for predicting class labels, almost all follow up work on representation learning using ViTs has followed this practice. Here, the role of [CLS] token is similar and is typically used for evaluation of learnt representation via linear probing. MAEs also included [CLS] token, likely to maintain consistency with existing literature and evaluations. In SiamMAE, much like in MAEs, the [CLS] token doesn't play any role during pre-training. This design choice was inherited from both ViT and MAE. Our ablation study shows that while the [CLS] token isn't crucial for achieving good results, omitting it slightly hurts the performance.
>
> | [CLS] token  | J & Fm    |
> | ------------ | --------- |
> |              | 57.5      |
> | &#10003;     | **58.1**  |
>
> >*General principle/strategy for preventing shortcut learning*
>
> We agree that a general strategy of preventing shortcut learning would be great and is indeed an open question in the field of representation learning for computer vision.
>
> >*Impact of dataset similarity has not been investigated*
>
> We agree that the role of dataset similarity between the train and test tasks has not been studied and will add this limitation in the text. However, we would like to point out that most of the prior work we compare with (UVC, VFS, MAE-ST and VideoMAE) were trained on the same dataset i.e. Kinetics-400.

---

> > ### Comment · Reviewer_e5XC · 2023-08-21
> >
> > > Confidence intervals
> >
> > I was referring to CIs for the distribution induced by the random variables in the training procedure (shuffling of training set, initialisation of model parameters), not inference. It would be best if these could be obtained from at least 3 trials for the final version.
> >
> > > [CLS] token
> >
> > Thank you for running the experiment without a [CLS] token.
> >
> > I understand that the [CLS] token may be useful in intermediate layers as a kind of position-agnostic "place" to accumulate global information, but it's a little unclear to me why the attention maps for the final layer would be meaningful if no loss is applied to the [CLS] token (Section 4.4). Is it possible to shed some light on this? Perhaps because the other tokens "use" the global information without necessarily "adding to it" in the final layer?
> >
> > This reminds me - are the query and key projections symmetric (i.e. $W_Q = W_K$) in the self-attention layers?
> >
> > > Impact of dataset similarity not investigated
> >
> > Thank you for acknowledging this.
> >
> > > Larger models
> >
> > Thank you for running initial experiments.
> >
> > > Future vs past
> >
> > Thank you for running this experiment; I hope it can be included in the final version.
> >
> > **Overall**
> >
> > The paper is intuitive, well-motivated and well-written. The authors have used the rebuttal to strengthen the paper. I do not have any major concerns and keep my initial positive rating.

---

> > > ### Author Response · Authors · 2023-08-21
> > > **Response to Reviewer Comment**
> > >
> > > Thank you for your comments and suggestions. We address the reviewer concerns below:
> > >
> > > >Confidence intervals
> > >
> > > Ideally, we would like to run all our experiments/ablations multiple times. However, we note that this is not a common practice in representation learning literature due high compute requirements per experiment. Nevertheless, we will re-run our base ablation setting 3 times to quantify the variance (if any) and include it in the revised version.
> > >
> > > >[CLS] token
> > >
> > > We don’t have a good understanding of why the attention maps for the [CLS] token are meaningful. Exploring this is an interesting research question which we leave to future work.
> > >
> > > >Are the query and key projections symmetric?
> > >
> > > No, we don’t use the same query and key weights.

---

### Author Rebuttal · Authors · 2023-08-09

We sincerely thank the reviewers for their thoughtful and constructive feedback. We are glad that **all** the reviewers found our method simple, clever and intuitive, our analysis and ablations to be thorough and convincing, and results impressive. The main aim of this rebuttal is to improve this work further by incorporating reviewer suggestions and comments. Specifically, we have conducted all additional ablations and experiments suggested by the reviewers. Here we comment on the feasibility and scope of two suggested avenues of improvement and provide some encouraging preliminary results.

>*Training larger backbones like ViT-B & ViT-L*

We note that ViT-S has approximately the same number of parameters as ResNet-50, enabling us to compare our method across a diverse set of baselines. Moreover, ResNet-50 is the largest backbone explored by prior work [12, 14, 16], all of which aimed at improving representations for correspondence. However, we agree that a systematic investigation into the scalability of our method in terms of model size (refer to L297) would be valuable. Practically speaking, we are constrained by resources. For instance, training a ViT-S/8 model on 8 Titan RTX GPUs for 2000 epochs requires approximately 16 days. Given the compute and time constraints, we trained a ViT-B/16 model for 400 epochs.

| Model      | J & Fm    |
| ---------- | --------- |
| ViT-S/16   | 58.1      |
| ViT-B/16   | 58.6      |

The improvement in performance is encouraging. To contextualize the magnitude of the improvement, we note that scaling DINO from ViT-S/16 to ViT-B/16 results in an improvement of 0.5 J&Fm.

>*Evaluation on video action recognition and image recognition tasks*

We agree with the general sentiment of the comment, emphasizing the evaluation of a self-supervised representation learning method across a wide array of tasks. While we aim to demonstrate the versatility of our method, following prior work [12, 14, 16], we've limited our claims and experiments to validate our method's effectiveness at learning representations for visual correspondence. A key advantage of current evaluation strategy is its computational efficiency as it requires no training. Evaluation on video recognition benchmarks often involves fine tuning models for 300 epochs (as seen in VideoMAE, MAE-ST), requiring approximately 32-64 industrial-grade GPUs (such as V100, A100).

---

### Decision · Program_Chairs · 2023-09-21

**Decision:**

Accept (oral)

**Comment:**

All reviewers find the paper to be novel with impressive experiment results. The rebuttal additionally resolves minor concerns regarding technical details and additional experimental comparisons. The authors are suggested to incorporate constructive feedbacks from the review comments into the finalized version. Based on high quality of this paper and the potential impact to the community, reviewers generally nominated it to be an Oral.